# Review of Livestock Welfare Indicators Relevant for the Australian Live Export Industry

**DOI:** 10.3390/ani10071236

**Published:** 2020-07-21

**Authors:** Patricia A. Fleming, Sarah L. Wickham, Emma J. Dunston-Clarke, Renee S. Willis, Anne L. Barnes, David W. Miller, Teresa Collins

**Affiliations:** College of Science, Health, Engineering and Education, Murdoch University, Perth, WA 6150, Australia; sarahlwickham@outlook.com (S.L.W.); Emma.Dunston@murdoch.edu.au (E.J.D.-C.); reneeswillis@gmail.com (R.S.W.); A.Barnes@murdoch.edu.au (A.L.B.); d.miller@murdoch.edu.au (D.W.M.); T.Collins@murdoch.edu.au (T.C.)

**Keywords:** physiology, behaviour, quality assurance, welfare indicators, benchmarking

## Abstract

**Simple Summary:**

The live animal export industry is under increasing public scrutiny to ensure optimal animal welfare conditions are met. To date, the main animal-based welfare indicator used to review and regulate this industry has been mortality. For a proactive industry wanting to transition to reporting on animal welfare not mortality, a broader suite of measures is needed. We reviewed a total of 71 potential animal welfare measures, identifying those measures that would be appropriate for use throughout the live export chain for feeder and slaughter livestock animals, and categorised these as animal-, environment- and resource-based. After considering the industry-specific constraints for animal welfare assessments, measures were categorised according to their application to the three identified sectors of the live export industry. We identified measures already undertaken or that are relevant for specific situations as part of the industry. Further research is currently underway to validate the applicability and value of these measures.

**Abstract:**

Animal welfare is an important issue for the live export industry (LEI), in terms of economic returns, community attitudes and international socio-political relations. Mortality has traditionally been the main welfare measure recorded within the LEI; however, high mortality incidents are usually acted upon after adverse events occur, reducing the scope for proactive welfare enhancement. We reviewed 71 potential animal welfare measures, identifying those measures that would be appropriate for use throughout the LEI for feeder and slaughter livestock species, and categorised these as animal-, environment- and resource-based. We divided the live export supply chain into three sectors: (1) Australian facilities, (2) vessel and (3) destination country facilities. After reviewing the relevant regulations for each sector of the industry, we identified 38 (sector 1), 35 (sector 2) and 26 (sector 3) measures already being collected under current practice. These could be used to form a ‘welfare information dashboard’: a LEI-specific online interface for collecting data that could contribute towards standardised industry reporting. We identified another 20, 25 and 28 measures that are relevant to each LEI sector (sectors 1, 2, 3, respectively), and that could be developed and integrated into a benchmarking system in the future.

## 1. Introduction

Over recent years, there has been an increase in concern from the general community regarding animal welfare in livestock production [1,2]. ‘Social licence’—whether a sector has ongoing approval and broad acceptance within the local community and other stakeholders—is vital for the continued support of all livestock production industries. Such concerns particularly impact the live export industry (LEI), where there is a divide between community expectation, government regulation and LEI performance [3]. Heightened public awareness around the industry (e.g., [4]) means that improving welfare outcomes and avoiding high mortality situations are priorities. 

Developing comprehensive measures of animal welfare is a necessity for quantifying and communicating outcomes. Quantifying and monitoring animal welfare (Box 1) across the entire live export supply chain is therefore an important step towards reporting and transparent quality assurance, which can be used to educate and reassure the general community. Most importantly, quantifying animal welfare as part of the LEI will enable future benchmarking—establishing criteria that can become aspirational and encourage continuous industry improvement.

In this review, we examine why animal welfare assessment is needed for the LEI, examining the current industry and legislative context under which animal welfare is currently managed and regulated. Next, we consider who should carry out animal welfare assessments, as well as potential constraints of assessment under industry conditions. The aim of this review was to identify internationally accepted and currently used indicators of animal welfare relevant to the LEI. Specifically, we sought to identify relevant indicators for each point in the live export supply chain for feeder and slaughter cattle and sheep that could contribute to developing a benchmarking system for animal welfare using a ‘Welfare Dashboard’ [5]. We have therefore reviewed a range of potential animal welfare measures, identifying (I) those that are already undertaken by the LEI, (II) those animal welfare measures that are relevant to the industry but are not required to be recorded under current Australian Standards for the Export of Livestock (ASEL) v2.3 (2011) regulations and (III) those that have limited application for day-to-day management but are more likely to be used for specific situations (e.g., research, sentinel animals). These findings are summarised in a table listing 71 potential animal welfare measures that address the 12 welfare criteria and 4 welfare principles identified by the European Union’s Welfare Quality audit. We detail these measures in the Appendix A accompanying this publication. Our conclusions include identification of potential future direction. 

Box 1What is animal welfare?In order to begin to measure animal welfare, we need to first understand what it is. There is a growing body of literature that examines the definition of what good animal welfare means (e.g., [2,6,7,8,9]). Animal welfare is a multidimensional concept that embraces the physical and mental aspects of the animal, including positive, neutral and negative mental states [10,11], the animal’s physiology and functioning, as well as its interaction with the surrounding environment and how the animal copes with the challenges faced. Many measures of welfare are needed because stressors can act on one or several parameters at different times and to differing degrees [12]. For example, an animal experiencing thermal challenge on a hot day will feel uncomfortable and show physiological signs of hyperthermia, will consequently alter its behaviour to seek shade, while the need to perform this behaviour may, in turn, result in a lowered function. Different animals may exhibit varied responses to the same challenge, due to genetics, sex, body weight, level of acclimatisation, etc., which can be addressed when multiple welfare measures are collected, allowing for some interpretation as to when normal responses to stressors become aberrant or pathological.Mellor and colleagues [6,13] have proposed Five Domains of animal welfare: four physical/functional domains which all impact on the fifth domain, that of mental state. The European Union Welfare Quality^®^ program similarly recognises four key welfare principles and 12 welfare criteria for animal welfare assessment [14]. We have used these Domains and Principles (Table 1) as a framework for understanding animal welfare for the current review.

**Table 1 animals-10-01236-t001:** The Five Domains of animal welfare [7] (left hand column) and their association with the Principles (middle column) and Criteria (right hand column) forming the basis of the Welfare Quality^®^ assessment [14] protocols. Colours are cross-referenced to Table 3.

Five Domains	Welfare Principles	Welfare Criteria	
1. Nutrition	Good feeding	1	Absence of prolonged hunger
		2	Absence of prolonged thirst
2. Environment	Good housing	3	Comfort around resting
		4	Thermal comfort
		5	Ease of movement
3. Health	Good health	6	Absence of injuries
		7	Absence of disease
		8	Absence of pain induced by management procedures
4. Behaviour	Appropriate behaviour	9	Expression of social behaviours
		10	Expression of other behaviours
		11	Good human-animal relationship
5. Mental state		12	Positive emotional state

### 1.1. Why Do We Need to Measure Animal Welfare in the LEI?

Australia is a major exporter of livestock and the industry is worth $1.79 billion to the Australian economy [15]. In 2018/9, almost 10,000 people were employed across the live cattle supply chain, which exported a total of 1.26 million cattle (89% of which were feeder/slaughter animals) worth AUD 1.64 billion Free On Board value) [15]. Furthermore, in the same year, 0.989 million sheep worth AUD 142 million, and 18,650 goats worth AUD 7.2 million were exported live from Australia [15] (Figure 1). The sheer number of animals that are processed through the LEI warrants the development of animal welfare assessment tools that are tailored to the conditions experienced throughout the supply chain. 

Most cattle and sheep are transported by sea, principally to south-east Asia and the Middle East (cattle and sheep respectively), while goats are largely transported by air to south-east Asia. Because most animals are transported by sea, most animal welfare measures have addressed this form of transport; much less has been done about animal welfare as part of air freight [16]. 

Currently, the Australian LEI and stakeholders use on-board mortality and compliance with the Australian Standards for the Export of Livestock (ASEL; [17]) and Exporter Supply Chain Assurance System (ESCAS; [18]) as indicators of welfare for feeder and slaughter animals. LEI industry reviews have previously provided suggestions that would lead to improved animal welfare monitoring. For example, the 2011 Farmer Review [19] recommended that the adoption of a Quality Assurance system that included all sectors of the supply chain would provide the needed ‘whole of industry’ surveillance approach, while also complementing current government regulatory compliance programs [20]. More recently, the 2018 McCarthy Review and 2018 ASEL review recommended that the LEI cease relying on mortality as the primary indicator of animal welfare. This approach is supported by stakeholders and the industry’s desire to show best practice is applied to avoid adverse publicity [21]. 

The issue with simply monitoring mortality and non-compliance with regulation is that problems can only be indicated retrospectively. Under current animal welfare reporting, the main welfare issues identified for the LEI are environmental conditions, resource access, and species-specific causes of morbidity and mortality. Being able to effect change proactively, on the basis of animal outcomes that do not result in mortality, could enable pre-emptive modifications and adaptive management solutions before animal welfare is compromised. Furthermore, a single incidence of high mortality substantially contributes towards negative community attitudes to live export, while continued efforts made by the industry towards managing animal welfare along the supply chain can go unrecognised. 

Previous reviews have been informative for the development of animal welfare measures as part of the LEI. In 2004, an Australian study identified seven key indicators of welfare on-board a ship: mortality, clinical disease, respiration rate, wet bulb temperature, space allowance, change in body weight, and ammonia levels [22]. For many measures, baseline information from which deviance from optimum and/or critical levels could be detected, has not yet been established. Application of welfare measures across the whole supply chain (including on trucks, in feedlots, and in abattoirs both in Australia and at the importing country) would require other indicators that can be applied more broadly. Behavioural indicators were not identified as part of the 2004 study, although understanding and application of behavioural measurements to assess welfare have markedly advanced over the intervening decade [23,24,25]. A 2015 review described 19 animal-based indicators considered valid for assessing sheep welfare and of these nine were considered feasible for use in UK abattoirs [26]. These indicators were: body cleanliness, carcass bruising, diarrhoea, skin lesions, skin irritation, castration, ear notching, tail docking, and ‘obviously sick’ animals. Grandin [27] provided five animal welfare measures for auditing purposes at stunning in the USA; the efficiency of stunning, use of electric prodders, and cattle behaviour during handling and the procedure (vocalisations and falls). An aim of this study was to expand on these published reviews and address the specific context of the LEI.

### 1.2. Industry and Legislative Context

To understand the complex legislative context of the Australian LEI, we reviewed all current standards and regulations pertaining to livestock transport and slaughter at International (2 sources) Commonwealth (8 sources), State and Territory (10 sources) levels. We also located five LEI reviews that have occurred since 2003 in addition to two national animal welfare and health schemes and five Australian livestock industry-led animal management schemes and programs (Table 2). These sources were obtained from searching Australian Government and livestock industry websites, which provide details on the standards, regulations and programs. Sources that were not relevant to livestock management on farms, in feedlots and/or during transport (both land and air) were not included in the review. 

In Australia, there is a broad trend in animal welfare regulatory reform to reduce prescriptive regulation and to move to a model of shared-responsibility with non-government stakeholders (industry, community) playing more direct roles. This is described in general government information related to good regulatory practice and in current Commonwealth and state regulatory reforms, as illustrated through various Biosecurity Bills/Acts/Standards and Guidelines at Commonwealth and State levels (Table 2). 

Animal welfare in live export is a complex regulatory issue within Australia. Exports are within the domain of the Commonwealth, but animal welfare and disease control are regulated at State/Territory levels (Table 2). In some States, there is further delegation of responsibility to other bodies (RSPCA inspectors, State Government Departments etc.). This hierarchy of laws and regulations contributes to conflicts and lack of clarity over roles, responsibilities and relevant legislative instruments [20].

The development of the Livestock Global Assurance Program (LGAP) is an important component of a shared-responsibility model [28]. LGAP follows a number of international standards, such as the World Organisation for Animal Health (OIE), Exporter Supply Chain Assurance System (ESCAS) and World Trade Organisation (WTO). AniMark is an independent, not-for-profit Australian company that has been appointed to implement LGAP [15]. Under current ESCAS policy, exporters are responsible for appointing auditors who report to government via the exporters. Under LGAP, facilities and operators will perform internal audits to be prepared for external auditing, with AniMark to train and appoint auditors who will review and report on all levels of the chain directly back to LGAP. It is likely that new proposed LEI welfare indicators would be valuable and could to be implemented through the proposed LGAP system.

## 2. Constraints on Animal Welfare Assessments

Conducting welfare assessments across all stages of the export chain (1. Australian facilities, 2. Vessel and 3. Destination country facilities) may be constrained by the vast environmental and management differences animals are exposed to. Subsequently, welfare assessment measures need to be specific to, or sensitive to, changes across LEI sectors. Here we discuss who should carry out welfare assessments and possible physical and logistical constraints on recording as part of consideration of applicability for the LEI.

### 2.1. Who Should Carry Out Welfare Assessments?

As part of developing animal welfare assessment methods for the LEI, consideration should be given to who will be undertaking the observations and how long such assessments would take. The frequency and duration of the assessments will also influence the credibility of the reporting. Welfare checks are something that AAVs (Australian Accredited Veterinarian) and proficient stock handlers perform when working with livestock on a day-to-day basis [41] to ensure the management of the livestock. A Welfare Dashboard [5] could be used to capture standardised reporting of the information observed during these daily assessments. Alternatively, comprehensive welfare measures could be included in existing pre-export inspections, shipboard reporting requirements and ESCAS audits. 

A survey conducted in 2015 indicated that the general public and animal welfare advocates strongly favour assessments of animal welfare as part of the LEI being carried out by independent animal welfare assessors [21]; ensuring quality assurance and confidence in the value of the assessments may rely on such independence. Hence, careful consideration of the review period and the selection of assessors with suitable auditing and inspection skills to report on livestock welfare is therefore required for each stage of the supply chain [42].

Current LEI reporting structures during sea voyages require the AAV or an accredited stockperson to provide Daily Voyage Reports for voyages ≥10 days, in addition to an End of Voyage Report. These reports and the End of Air Transport Journey Report are provided to the Australian Department of Agriculture, Water and the Environment (the Department) and exporter [17]. From October 2018, Independent Observers (IOs) were required to travel on all sea voyages, unless decided otherwise by the Department [43] (2019: IOs no longer required for all short-haul shipments considered ‘low-risk’ [44]), to observe and collect information on exported livestock from pre-export facility until the end of discharge in the importing country. IOs provide a report to the Department independent of those provided by the AAV or accredited stockperson [45]. Exactly who would be responsible for the collection of additional animal welfare measures and the standardisation of collected data during sea voyages should be determined by the industry and the Department.

### 2.2. Constraints to Animal Welfare Assessment Protocols

#### 2.2.1. Individual vs. Group Assessments

On board and in pre- and post-vessel facilities, livestock are usually housed in close proximity; stock handlers moving around animals/pens can cause disturbance that may confound some measurements being recorded. Thus, it is difficult to conduct a physical examination on any individual animal (e.g., temperature, auscultation, palpation, inspection of mucous membranes, collection of samples) because capture and restraint agitates all animals in the pen, causing unnecessary stress and the possibility of injury. Individual animals with ill-health may be moved to designated hospital pens where they can be restrained for individual animal examination and treatment; however, collecting physiological information from these isolated sick individuals will not necessarily provide representative physiological information for the animals in the home pen. Consequently, it is suggested that most of the currently proposed welfare measures would be taken by observation and collected at a group (pen) level without removing individuals [5].

#### 2.2.2. Visibility

Animal welfare assessments require good visibility of animals. The use of yards, races and crushes to move and restrain animals, allowing handling, examination and treatment, varies between countries and livestock species. Cattle in Asian countries tend to be handled individually with varying or little use of yards etc, while sheep in Middle Eastern countries may be managed in yards/feedlots that are similar to those in Australia. Where handling facilities are used for holding and lairage, lighting, the availability of equipment, and the age, condition and hygiene of these facilities can vary markedly. There can also be a welfare cost to additional handling required for some animal welfare assessments.

During vessel transport itself, low light conditions may mean that a single vantage point has restricted visibility of some parts of the pen. Furthermore, ship pen design follows the structure of the decks, and therefore some parts of the pens may be obstructed from view. Stocking density will also preclude visibility of all parts of each animal and bedding can reduce visibility of animals’ feet. As part of air freight, animals are held in crates and access to the cargo hold may limit visibility *en route*. 

#### 2.2.3. Restricted Animal Movements During Transport

The restriction of movement by the animals due to pen size and stocking density will mean that behavioural differences (e.g., changes in gait) may not be obvious in some contexts. Under dense stocking conditions, such as on trucks or in crates, animals may have limited opportunities to move away. 

#### 2.2.4. Appropriate Times to Carry Out Assessment 

Experience, skills and stock handler skills are likely to vary greatly between sectors of the LEI supply chain, while there are also different attitudes towards livestock handling across the globe [46,47,48]. The staff and the time available to carry out animal welfare assessments will therefore vary across the chain and needs to be considered when designing protocols. For example, loading and unloading of vessels are carried out as efficiently as possible to reduce the time trucks wait at the port and ship docking times, resulting in all personnel being occupied. Animals are also likely to show heightened responses during loading and unloading and any handling procedures, which may mask more subtle behavioural or physical responses. However, recording animal gait, stock handler skills and the use of goads to move livestock on/off trucks and the ship are possible. An appropriate time for the initial assessment may be after penning (feedlot) or loading (vessel) is completed and the animals have had a chance to settle into their novel environment, with regular daily assessment required thereafter. Time of day, ambient temperature, feed routine and circadian rhythms need to be considered when selecting the optimal time of assessment. Ensuring that measures are appropriate to the context and environment will therefore need to be considered [5].

### 2.3. Review of Potential Animal Welfare Measures and Their Applicability to the LEI

To establish a list of LEI-applicable animal welfare measures, we reviewed all current legislation/regulations/guidelines relevant from on-farm through to market (e.g., [18,31,40]; full list in Table 2) to identify recommended and implemented animal welfare measures. Using search terms for animal welfare, transport, live export, handling, we also carried out a literature search of Australian (e.g., [22]) and international studies (e.g., [26,27]) that proposed measures applicable to livestock. In considering applicability and relevance to the LEI, we captured the requirements of each method (i.e., whether it required trained personnel or specialised equipment, and whether there were already threshold values for animal welfare expectations for each measure) and considerations relevant to the LEI (i.e., requires animal handling, involves invasive measures, the relative cost of the measure above operating costs, and approximate amount of time that would be required). 

Welfare indicators are currently being developed for shipboard reporting [5] that could be collected by AAVs, accredited stockpersons or IOs. These may also be incorporated as part of ASEL and ESCAS, or within LGAP audits in the future. Our review provided a list of 71 potential indicators of animal welfare (Table 3) and their application to all stages of the LEI. As the LEI involves many different environments, we divided the industry into three sectors: (1)Australian facilities, including farm-yards, pre-export facilities and land transport;(2)Vessel (ship and aeroplane); and(3)Destination facilities, including feedlots and abattoirs.

When considering the application of each of the 71 measures to these three sectors, the relevant legislation and regulations were reviewed to determine which measures are currently regulated or not. Specifically, Animal Welfare Acts (Western Australia, Norther Territory, Queensland and Victoria), Codes of Practice, Animal Welfare Standards and Guidelines [49,50,51,52] and the National Feedlots Accreditation Scheme ([53]; applicable to feedlots who wish to produce grain fed beef to domestic and international markets; note, not all pre-export facilities are NFAS accredited) for Australian facilities, ASEL for sea transport, and ESCAS auditing for destination facilities. We described the indicators in terms of animal-, environment-, resource-, and management-based measures (Table 3). Some of these measures are carried out at an individual animal level (e.g., body temperature), and therefore, a subsample of sentinel animals could be monitored as part of the LEI process. Most measures can be carried out at a group level (e.g., behavioural scores (posture, activity and demeanour) or respiratory panting scores in a pen of animals) [54,55,56,57]. Here we summarise the feasibility and application of each measure to the LEI. Supporting information for each measure is detailed in the Appendix A.

### 2.4. Measures that Are Already Undertaken by the LEI

#### 2.4.1. Management

It has been well established that stock handler attitudes and skills influence the way they handle livestock, which directly impacts an animal’s welfare. Many measures relating to industry management (e.g., livestock sourcing, rejections, animal traceability, electric prodder use; Table 3) are already being documented [61,62,63]. Recording certification of stock handler training in low stress handling, and experience etc. for truck drivers, feedlot workers, ship crew, and abattoir workers, for each consignment, has the potential to be included in the Welfare Dashboard.

Animal management substantially influences the welfare of the animals under consideration. Many of these measures are currently monitored through ASEL and ESCAS, for example sourcing, breed, mixing, rejections, traceability, isolation/separation (Table 3). Being able to capitalise on improvements in this area requires traceability of livestock through the LEI chain, which could be facilitated through better tracking facilitated by radio-frequency identification (RFID) tags and central collation of these data.

Some measures are currently undertaken within the LEI as part of day-to-day proceedings, e.g., as part of the Daily Voyage Report and End of Voyage Report requirements under ASEL [17,59], and through the regulation and auditing of industry sectors (NFAS, Animal Welfare Standards and Guidelines and ESCAS). Existing compliance requirements are therefore being used as the starting basis for the development of welfare indicators (e.g., LIVEX-Collect [64]), and some/many of these measures can be extended forward in the supply chain to be used during land transport and in feedlots and abattoirs in the destination country. The various resource, environment and management measures currently recorded as part of animal welfare assessments, predominately during the sea transport stage of the supply chain that have applicability to the whole LEI supply chain (Table 3) could be included in a Welfare Dashboard [5].

#### 2.4.2. Mortality

Mortality recorded in each sector of the LEI indicates the effect of overwhelming disease, injury or lack of care, and therefore the end result of poor quality of life for the animal [65]. Mortality is already recorded at all sectors of the LEI. During sea and air transport, mortality is reported by the AAV, accredited stockperson or exporter directly to the Department and is not accessible or used for any other purpose (except at the conclusion of investigation for incidences that exceed the reportable mortality rate for that species [66]; Figure 2). Daily mortality records should be included under the proposed Welfare Dashboard with more specific data detailing voyage day, location of events (e.g., decks) or class of animals, to inform and help manage future risks. Tracking animals through the live export chain (e.g., using National Livestock Identification System (NLIS) tags) will allow greater insight into morbidity (and mortality) events and is recommended as a first step towards developing this understanding of potential underlying causative factors. A tracking method could also allow the point of origin to be identified, which may start to identify patterns in on-farm conditions, such as animal exposure to feed types, vaccination histories etc. [67]. Such information could form part of risk assessments in the future. 

#### 2.4.3. Morbidity and Records of Animal Treatments

Disease and health status of animals is an important component of the overall status of the animals, and can reflect poor welfare [68]. There is also an economic cost to treating ill or injured animals and removing the animal from the export chain if required. A number of consignment delays and contentious situations can be traced back to incidence of disease [67,69], therefore, preventing the spread of disease and health issues has benefits for both animals and the industry.

Morbidity, records of animal treatments, and animals moved to hospital pens are important for the LEI as a method of monitoring health status and screening for potential disease outbreaks. Intensive housing with animals in close proximity, mixing of animals from a variety of origins, and increased shedding of organisms by animals under stress can result in the rapid spread of pathogens to previously naïve cohorts, so acute outbreaks of disease may be unavoidable. Therefore, evidence of active management responses towards disease or injury may be more relevant for a Welfare Dashboard than simply the incidence of disease. Daily observations of pens to identify individuals showing clinical signs of disease or injury are part of the routine of AAVs and stock handlers at all stages of the LEI. These could be expanded to include other measures of poor demeanour (e.g., dull, unresponsive) or abnormal behaviour, and animals classed as ‘obviously sick’, thus relevant to health status. Clinical signs that indicate poor welfare may include lameness, dyspnoea, coughing, nasal discharge, diarrhoea, ocular health, or excessive scratching or rubbing. These can all be assessed by observation at the pen level, such as the percentage of animals in the pen with each condition, without physically restraining the animals for individual examination.

#### 2.4.4. Environmental Conditions and Animal Responses

Environmental conditions can affect animal health and welfare. Periods of extreme heat are of particular relevance for the welfare of live animals exported from Australia across the Equator, especially when animals move from the southern hemisphere winter into the northern hemisphere summer [70]. 

Some environmental factors are already measured at some sectors of the industry. For example, during sea transport on voyages ≥ 10 days, daily records of (1) average dry bulb and wet bulb temperature for each deck, (2) humidity for each deck, and (3) bridge temperature (ambient) are reported under the Daily Voyage Report. A summary of environmental conditions (“comment on weather, temperature, humidity, ventilation and decks/bedding”) is required in the End of Voyage Report for all voyages. Australian feedlots are required to have a stationary weather station that constantly records temperature, humidity and solar radiation, facilitating the gathering of daily weather data. Continuous digital monitoring of temperature and humidity is now required on sheep voyages to the Middle East from June to September [17,70]. 

Heat stress reporting requirements do not currently extend beyond sea voyages ≥10 days unless a notifiable mortality limit is reached in a pre-export facility or during a short haul voyage. Collating environmental information as part of a Welfare Dashboard [5] would provide a robust method of monitoring and could assist in assessing the effects of duration and intensity of heat (or cold) on any group of animals for immediate and ongoing management, as well as determining site, vessel and seasonal trends, for future risk management.

Animal responses to their environmental conditions are also currently monitored to some extent; in particular the response of livestock to hot conditions. Sheep and cattle use the respiratory system as their principal means of heat dissipation, and progress through identifiable stages of panting with an increasing physiological response to the heat [71,72,73]. Panting scores are a quick, non-invasive indicator of respiratory character and hence welfare state that can be used in all areas of the LEI, which, in combination with assessments of behaviour and demeanour at a pen level, may reveal the extent of heat stress, as well as the presence of respiratory disease [74]. When combined with environmental or behavioural measures, the cause of panting can be determined, such as heat stress or fear response. Panting scores have been historically been assessed daily on a deck basis during voyages by sea as part of the Daily Voyage Report for voyages ≥10 days. Respiratory character (1 = normal, 2 = panting, 3 = gasping) and *“Whether and to what extent the livestock show heat stress”* are recorded. More detailed panting score measures have been included in Daily and End of Voyage reporting requirements for sheep during sea transport in response the to the McCarthy Review 2018 and the revision of ASEL 2018 [19,40]. Species-specific panting scores are provided in the veterinary handbook for cattle, sheep and goats [75], which is available to AAVs and stockpersons, facilitating the possibility of standardised reporting upon inclusion in the Welfare Dashboard.

#### 2.4.5. Feeding

More research is required to identify the effects of feed and water withdrawal, with novel welfare indicators needed to assess short term hunger and thirst [26]. Feed and water consumption and availability are part of the daily monitoring of livestock at pre-export facilities, the Daily Voyage Report (*“average per head”*), End of Voyage Reports (ASEL) and ESCAS auditing reports, although, there is no guideline for standardised collection of this information. For example, the ship reports require: *“Feed and water—comment on stock access and if there were any issues with maintenance”*, leaving reporting of this information to individual discretion. Maximum time off feed and water is regulated under Australian Standards and Guidelines for transport, under ASEL for sea transport, and at abattoirs under ESCAS. These measures have the potential to be expanded as part of the Welfare Dashboard, as developing guidelines for their recording would improve standardisation and repeatability of monitoring. 

### 2.5. Relevant Measures, Not Currently Required in the LEI, That Could Be Applicable to Welfare Assessments

#### 2.5.1. Management

Collating land transport details, such as time off water and food, and time between mustering, loading and transport (regulated under ASEL in Australia) along with any issues, including delays in transport and animal injury as the animals progress through the chain, could provide long term monitoring and indicate areas for improvement. Delays experienced at the wharf during loading or unloading of the livestock could be included here. The transmission of data about each cohort of animals along the chain could enable prospective management decisions that optimise the subsequent care and therefore welfare of those animals.

Appropriate design and construction of facilities can increase productivity and reduce welfare risks. The ESCAS audit process contains measures of facilities, such as presence of protrusions or gaps where animals may be injured or trapped, adequate fencing to provide restraint, width of races, non-slip flooring, and animal responses to facilities, including rate of animal slips and falls and the number of times animal flow stops. The use of the ESCAS welfare audit and ASEL regulations may be a starting point for developing standardised welfare assessments that are applied earlier in the supply chain, such as during loading and unloading at farm, pre-export facilities, ship/aircraft, destination feedlot/farm and abattoir. Hygiene indicators used at abattoirs need to be modified to enable hygiene assessments for transport vehicles and feedlots.

#### 2.5.2. Food and Water Access and Consumption

Measures that relate to both immediate and longer term availability of food and water and consequences of consumption can be made. These are fundamental to health and welfare of the animals, ensuring their needs are met, and are very important in the LEI due to the link of inappetence of sheep with disease such as salmonellosis [76,77] and the need for adequate hydration especially during hot environmental conditions so that physiological cooling mechanisms can continue [78,79].

Rumination and gut fill can be determined on observation of the animals as an indication of recent feed and/or water consumption [80,81], and are measures that can be used in day-to-day management. Rumination is best observed in undisturbed animals at rest, and as a group measure [82,83,84]. Gut fill could be assessed subjectively, by observing the animal for abdominal distension or hollow sides in the flanks behind the ribs; standardising this assessment can be aided by using image charts and a grading system from empty (hollow sides) to full [85,86]. 

The behaviour of animals at feeding could be used as a measure of how hungry the animals are, especially useful in situations where animals receive intermittent feeding, when they may demonstrate behaviour indicating urgency to eat. Animals that stay away from feed can also be detected, and may be animals suffering disease, or may indicate concerns related to adequacy of trough space and food for the group size and hierarchy. The responses of animals to provision of water, for example after troughs are cleaned or refilled, could indicate thirst, as well as a desire to access cooler water under hot environmental conditions [87].

Body mass and body condition score (BCS) can indicate the longer term results of feeding. ASEL state the minimum and maximum weight and BCS for animals entering the LEI, and monitoring and recording will allow compliance to be demonstrated. Change in BCS over time can reveal how animals are being managed and coping with their environment [88]. While it is usual practice during live export of sheep to feed only at maintenance, cattle may be fed for weight gain; loss of BCS over the process indicates lack of feeding. The BCS at any point in time may also provide information that can be used in risk assessment; for example animals in heavy body condition may be more susceptible to heat stress and to injury [66]. Sheep in high BCS may be more susceptible to metabolic effects of inappetence in the second half of the year [89], while heavy cattle may be less capable of dealing with rough seas, and may be more susceptible to foot and leg injuries. 

Alternatively, or in addition, carcass classification can provide information regarding long term feeding, with the possibility of video image analysis technologies being tested [90].

#### 2.5.3. Behaviour

Assessing behavioural events (behaviours that occur both frequently and infrequently) can be performed by visual observation, it is non-invasive and does not require specific equipment. For example, shivering could be recorded as an indicator of possible thermal discomfort, while specific abnormal behaviours, such as bouts of aggression or inappetence, could identify animals that may need to be removed from a pen for treatment. Stereotypical behaviour is common in confined animals [91] and generally indicates long-term challenges for animals, i.e., over months or longer [92] or can be a coping mechanism [93]. The reporting of stereotypical or unusual behaviour in confined animals [91] can be used to indicate the mental state. The expression of stereotypies as part of the LEI could reflect boredom (e.g., in feedlots and during sea transport) and environmental enrichment may reduce the expression of stereotypies. 

Activity budgets—capturing the proportion of time that an animal spends doing particular actions—provide an easy to measure, non-invasive method that can target behavioural states that are welfare-relevant (e.g., eating, resting). Although ethograms can be time consuming with large numbers of animals, developing monitoring protocols for representative groups is achievable. These measures have the potential to be expanded and used as part of the Welfare Dashboard [5].

The incidence of particular activities can also be informative. For example, evidence of panting can reveal how animals experience the temperature and humidity of their environment. Currently, panting score is monitored at pre-export facilities and reported daily under ASEL. If a standardised reporting format was presented, these data could be collated and compared with weather conditions and animal handling procedures, with the potential to reveal sectors of the LEI that pose the most risk for heat or cold stress conditions and respiratory disease. 

Observing behavioural preferences can also reveal how animals are responding to their environment. For example, shade is important for livestock in feedlots, especially in climates with high solar radiation and high temperatures, and sheep (e.g., [94]) and cattle (e.g., [95]) use shade if it is available. The effectiveness of shade types can be measured by observing animal outcomes (such as animal preferences and thermoregulation) under varying conditions and shade access [96]. 

Qualitative Behavioural Assessment (QBA) can indicate ‘how’ the animal is behaving rather than what it is doing, by looking at how the animal interacts with its environment [41,97]. QBA is a quantitative measure of the animal’s demeanour, capturing subtle differences in behavioural expression that reflect an animal’s environment as well as their physiological state [98,99,100,101,102]. Consequently, QBA scores are a useful measure of the animal’s experience with its environment and can reflect the valence of its emotional state. QBA was included as one of 13 measures as part of the 2004-2009 European Commission’s Welfare Quality^®^ audit [103,104,105] and is aligned with the scoring of demeanour as ‘bright, alert, responsive’, which is a routine tool for veterinary practice (e.g., [106]). QBA should be applicable for all species and in all areas of the LEI supply chain, although a degree of training is required for data collection, analysis and interpretation of results. Developing protocols can provide immediate feedback to the assessors, but the data can also contribute to long-term analyses. For example, current practice around sorting animals at loading and scanning pens as part of daily monitoring involves stock handlers observing animals and using their judgement to identify animals that behave differently; this informal approach is effectively using aspects of the animal’s demeanour to identify and sort individuals. A more formal approach could be developed using appropriate scoring sheets developed for particular species and stages, and results from this provide feedback to training packages for stock handlers. These measures have the potential to be included as part of the Welfare Dashboard and contribute to long-term datasets.

#### 2.5.4. Environment

Environmental measures, such as temperature and humidity, at pre-export feedlots and on vessels (both ships and aeroplanes) are already monitored in the LEI (Table 3). Ventilation serves to maintain environments appropriate to the physiological needs of livestock [107,108]. Ventilation is important to remove air pollutants (such as ammonia and carbon dioxide, and dust) and maintain air quality. Ventilation in feedlots and on trucks is usually by natural means, whereas on board ship, ventilation is by mechanical means. High ammonia concentrations can irritate the eyes causing conjunctivitis, and upper respiratory tract leading to coughing (particularly on hot days) and rapid breathing. The small airways of the lower respiratory tract become inflamed after exposure to ammonia [109]. Environmental conditions in Australia, on board ship, and in the Middle East and Asia are favourable for ammonia gas production. Already part of LEI management procedures and regulation, ensuring frequent changing of bedding and adequate ventilation can bring fresh air and remove ammonia gas [109]; however, low air turnover and ventilation dead spots can be issues on board ships [107].

#### 2.5.5. Other

Ideally, pregnant livestock do not enter the feeder/slaughter export supply chain; those that do are recorded under the pre-export facility monitoring or Daily Voyage Report (“Births and abortions including estimated stage of pregnancy”) and health and welfare of the livestock (“the number of livestock born, the number of abortions and estimated stage of pregnancy”) as part of the End of Voyage Report. The conditions during export are not conducive to good welfare of heavily pregnant animals or those giving birth and the offspring. Currently, actions such as pregnancy testing and careful animal selection are taken to reduce the likelihood of pregnant livestock entering the feeder/slaughter LEI supply chain, with incidences of births and abortions during voyages collected under current reporting requirements. Collating this information as part of a Welfare Dashboard would be advised for feedback to suppliers and subsequent improved management.

### 2.6. Measures Relevant for Specific Situations (e.g., Research, Sentinel Animals), but Unlikely to Be Used for Day-to-Day Management

We recognise that there are numerous animal welfare measures proposed for on-farm protocols that are unlikely to be feasible under normal LEI practice due to impracticality of the measurement (e.g., heart rate, body temperature, stress hormones), where the measure recorded is not relevant to the short time frame within the supply chain (e.g., reproductive rate), or measures are delayed and therefore would not provide immediate feedback required to act immediately (e.g., meat quality).

Heart rate and heart rate variability are indicators of the emotional responses of an individual to a short-term problem, and can increase in anticipation to, and during, an event [110,111,112], changing within 1 or 2 heart beats [113]. This means that both measures can also be affected by the act of measuring them. Remote methods for monitoring may be useful in controlled situations; however, application under LEI is impractical.

Body temperature is an indicator of the onset or degree of thermal stress in an animal [114], and can also be used to indicate the presence of disease (pyrexia) and stress [115]. Body temperature of an animal can be measured rectally at a specific time point with a standard thermometer; however multiple readings of the same animal, in order to account for circadian patterns [116], require repeated handling. Remote continuous temperature monitors are available but are expensive (limiting the number of animals monitored) and can be intrusive. Body temperature is not a practical animal-based measure for the LEI with its use limited to necropsy, sick animals, those that are individually examined, and those used in research (e.g., [117]).

‘Stress hormones’, such as cortisol (e.g., [118,119,120,121]) as well as haematology and blood biochemistry (including acid-base disturbance; e.g., [118,121,122,123,124]), are measures that require blood sampling for assessment. Animals must be restrained to collect a sample, which is then sent away for analysis. This can be costly and may not provide immediate results. Additionally, other measures need to be used in conjunction to provide enough information for correct interpretation. Relevance in the LEI is, therefore, limited.

The assessment of carcasses for bruising and meat quality at slaughter may provide information on the recent experience of livestock. Glycogen is required for good meat quality, but glycogen is also the first energy store to be depleted, especially for stressed animals [125,126,127,128]. Stress therefore leads to a decline in acidification of the meat, which increases risk of spoilage and an abnormal colour that makes it difficult to market [129]. Consequently, meat quality can decline with both physical and emotional stress the animals are exposed to (e.g., during transport or from mixing unfamiliar animals together, or handling prior to slaughter [128,130,131]). Meat quality measures are usually obtained post-mortem, so provide only retrospective information for the animals tested. To collect these measures, access to the destination market abattoir would be required, while abattoir surveillance for disease and carcase condemnation could provide data on the health of the animals at slaughter.

## 3. Conclusions

Animal welfare assessment is required to identify compliance with legislation, policy and regulatory standards, market assurance, for the management of risks and in response to public attitudes and concerns [2,132]. The welfare status of animals also influences the quality of the product, either directly or indirectly, via consumer perceptions [133]. Proactive animal welfare monitoring and engagement with all stakeholders are therefore needed to ensure continued social licence to trade for the LEI.

Current LEI welfare assessments are focussed around mortality, morbidity, environmental measures and the resources provided to animals. This emphasis is likely to change as the government seeks to move towards more focus on reporting on animal-based measures of welfare that are more likely to be met through a combination of measures addressing ‘good feeding’, ‘good housing’, ‘good health’ and ‘appropriate behaviour’ (Table 1). We considered 71 potential animal welfare measures that address 12 welfare criteria and 4 welfare principles of animal welfare (Table 3). We identified the measures that would be appropriate for use as part of the live export supply chain, and categorised these as animal-, environment- and resource-based. After reviewing the relevant regulations for each sector of the industry, we identified measures already collected under current practice that can be expanded on to form a Welfare Dashboard: a LEI-specific online interface for collecting of data that could contribute towards benchmarking the industry [5]. Importantly, this suite of measurements will allow us to start identifying patterns or associations allowing algorithms to be developed that could rely on measuring a few key indicators to benchmark animal welfare. Although we note that many measures may be perceived by the general public as relevant to the LEI [21], we identified and dismissed measures that were not appropriate for the LEI due to impracticality.

Care is needed for compliance approaches based on environment-, resource- and animal-based measures, and tick-the-box assessments (i.e., using threshold values), since these are not necessarily associated with good welfare outcomes [42]. For example, mortality reporting uses fixed thresholds that trigger a formal mortality investigation [66]; however, not reaching mortality thresholds does not necessarily mean the majority of livestock experienced acceptable welfare outcomes. Increasing the scope of animal welfare assessment tools could therefore extend the capacity to drive improvement.

The need for effective feedback and continuous improvement on performance requires established and detailed protocols for consistency over time and locations and between practices. Monitoring can be useful for exporters to measure the performance of a facility, supply chain or management team, such as by differentiating between average and high performers, or detecting declines in performance before actual non-compliance occurs [134]. We note that awareness of the differing objectives of assurance systems compared with benchmarking is important. While assurance systems work towards compliance, they do not naturally engender self-driven improvements. In the case of the LEI, benchmarking is likely to increase ownership and investment into the system, with the LEI participants determining how they best modify their own systems to improve welfare outcomes.

Engaging a range of novel sensors in ongoing monitoring could be productive in animal welfare assessments. Current technology is available to monitor animal physiology, including heart rate, core temperature (iButtons or loggers [135,136]) and surface temperature (thermal imaging cameras [137]), health (biosensors [138]), activity and posture (accelerometers and pedometers [139,140]) as well as bioacoustics [141,142]. Devices that can be externally mounted are desirable for minimised impact on the animal, although there are issues with damage in high density environments (e.g., animals chewing and damaging devices) such as part of the LEI. As these technologies develop and become more feasible and practical, adoption within the LEI may occur, although this will only progress if validity, representativeness and value of such monitoring is demonstrated. 

Environmental sensors that monitor temperature, humidity ammonia and carbon dioxide are also becoming more available and affordable [143]. For the adoption of these technologies to occur on vessels (ships and aeroplanes), many environmental constraints such as vessel design and infrastructure impeding signals of remote technologies need to be considered. Future LEI benchmarking systems could facilitate adoption of sensor systems as they are shown to be relevant and of value. 

Further work is required to develop a system that can be tailored to the logistics and requirements of the Australian LEI. Ideally, the aim would be to provide non-invasive, cost effective, and implementable measures that incorporate animal-based factors, such as behaviour and physiology, and (independently) environmental-based factors relevant to sheep, cattle, buffalo and goats. As well as capturing measures in such a way that LEI participants can use these data to improve their practice. Development of appropriate data entry forms (possibly through handheld devices) that feed into a web-based database can facilitate easy pen-side data collection of standardised information and make reporting requirements more efficient and effective. Collating and recording these data over a period of time can value add to current practice, providing the baseline data against which industry improvements can be measured.

Some areas are noted where more research is required. For example, the development of measures suitable to determine whether animals in the LEI have a positive emotional state, addressing potential time delays between entering data and accessing comparison with benchmarking data, as well as developing versatile and informative tools to communicate findings to a broad range of stakeholders. Identifying how various stakeholders want to see the industry transparently assessed [21] is also a consideration in building such tools. The next generation of industry regulation, and adaptation by the industry of new benchmarking and audit methods, will work towards addressing community expectations.

## Figures and Tables

**Figure 1 animals-10-01236-f001:**
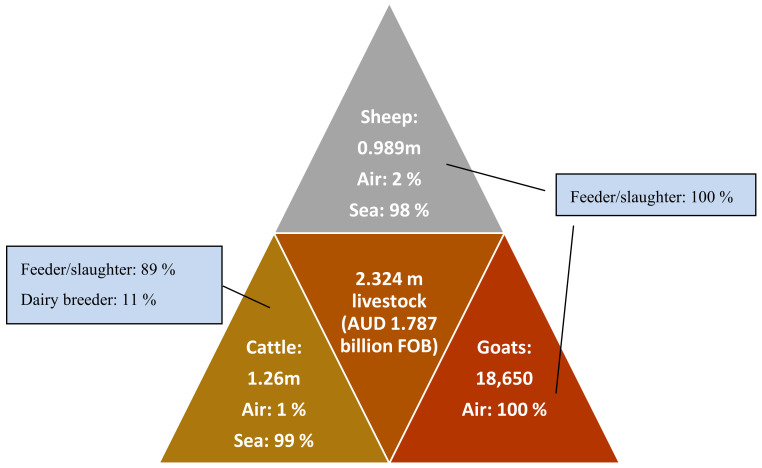
Animal numbers for live export from Australia for 2018/9 [15] by species (sheep, cattle, goats) and transport (air or sea), and total Free on Board (FOB)—value at the time of selling/buying the animals in Australia (i.e., the values quoted here do not include the cost of shipping/transport).

**Figure 2 animals-10-01236-f002:** Main causes of mortality for livestock during sea and air transport for 23 notifiable incidences (2012–2016) where the level of mortality on a consignment exceeded the reportable mortality rate (reviewed under Investigations into Mortalities carried out by the Department of Agriculture, Water and Environment [66]). Left are proximate causes of mortality identified by these investigations; those underlined were recorded for three or more of these investigations. Right are examples of contributing factors.

**Table 2 animals-10-01236-t002:** Laws or regulations that are relevant to the live export industry. Included are examples of government and industry-led management guidelines.

Jurisdiction		Purpose
International	International Air Transport Association (IATA) Live Animal Regulations (LAR) [29]	Global standards and guidelines to transporting animals by air in a safe, humane and in a cost-effective manner
	World Organisation for Animal Health (OIE) standards [30]	International trade standards for the transport and slaughter of animals produced for consumption
Commonwealth	Navigation Act 2012	Regulate international shipping
	Australian Maritime Safety Authority Marine Order 43 (AMSA MO43)	Cargo and cargo handling—livestock
	Australian Meat and Livestock industry Act 1997	Regulate industries
	Australian Meat and Livestock industry (Export Licensing) Regulations 1998	Regulate industries
	The Australian Standards for the Export of Livestock (ASEL) [31]	Standardise export procedures
	Exporter Supply Chain Assurance System (ESCAS) [18]	Regulate supply chain—requires evidence that animals will be handled and processed in accordance with the internationally-accepted OIE animal welfare guidelines
	Export Control Act 1982 and Export Control (Animals) Order 2004	Regulate animal export industries
	Inspector-General of Live Animal Exports Bill 2019 [32]	Overseeing monitoring and reporting, permits and licencing
State/territory	National Land Transport Standards [33]	Guidelines for land transport of livestock
State	Animal Welfare Acts	Overarching animal welfare legislation
	Codes of Practice/ Animal Welfare Standards and Guidelines [34]	State-level codes
Industry-led examples	Livestock Global Assurance Program (LGAP)	Overarching review and regulation of the LEI
	Livestock Production Assurance program [35]	
	Grazing Best Management Practices (Grazing BMP) [36]	Identifying best management practices
	National Dairy Industry Animal Welfare Strategy [37]	Leadership to improve animal welfare
	National Feedlot Accreditation Scheme [38]	Quality System for beef feedlots
Other	2003 Keniry Review	8 recommendations; Australian livestock export code established (which became ASEL)
	2011 Farmer Review [20]	14 recommendations; conditions around supply chains
	2018 McCarthy Review [19]	23 recommendations; including assessment of animal welfare rather than mortality, stocking rates, reportable mortality levels
	2018 Moss Review [39]	31 recommendations; including around internal regulatory practice, performance and culture, developing welfare indicators and using these as part of the regulatory framework
	2018 Technical Advisory Review of ASEL by sea [40]	49 recommendations, including stocking density, voyage reporting and on-board personnel and assessment of animal welfare
	Australian Animal Welfare Strategy (AAWS)	Information and development of future directions for improvements in animal welfare—no longer funded
	Animal Health Australia (AHA) Standards and Guidelines	Coordinating the development of national livestock welfare standards and guidelines

**Table 3 animals-10-01236-t003:** Mapping of animal welfare measures onto the European Union’s Welfare Quality^®^ framework, per sector of the livestock export supply chain (1 = Australian farm, pre-export feedlots and land transport, 2 = Australian Standards for the Export of Livestock (ASEL) (on-board the ship/plane reports), 3 = Exporter Supply Chain Assurance System (ESCAS) (destination feedlot and abattoir). Measures are identified as (C) currently used as part of the life export industry (LEI) in some form, (R) relevant to the LEI, (I) largely irrelevant to the LEI sector due to impracticality or lack of opportunity. Colours are cross-referenced to Table 1.

Welfare Principles		Welfare Criteria	LEI Sector			Animal-Based	LEI Sector			Environmental-Based	LEI Sector			Resource-Based
			1	2	3		1	2	3		1	2	3	
Good feeding	1	Absence of prolonged hunger	R	R	R	Digestion					C	C	C	Feed access
			C	R	R	Weight and body condition score (BSC)					C	C	R	Feed consumption
											C	C	C	Feed hygiene
											C	R	R	Time off feed
											R	R	R	Time to resume feeding
	2	Absence of prolonged thirst	I	I	I	Acid-based disturbances Hormones					C	C	C	Water access
			I	I	I						R	C	R	Water consumption
											C	C	C	Water hygiene
											C	R	R	Time off water
											R	R	R	Time to resume drinking
Good housing	3	Comfort around resting	R	R	R	Ethograms					R	C	C	Cleanliness, dry lying area at all times
			R	R	R	QBA ^a^								
	4	Thermal comfort	I	I	I	Respiration rate	C	C	R	Shade and shelter	C	C	R	Manure pad moisture
			R	C	R	Panting					R	R	R	Enough shade/shelter for all animals to access
			I	I	I	Body temperature								
	5	Ease of movement	R	R	R	Ethograms	C	C	C	Space allowance	C	C	C	Adequate space Flooring/terrain
			R	R	R	QBA					I	R	R	
		Other					I	C	I	Ventilation				
							I	R	R	Ammonia				
							R	C	R	Temperature & humidity				
							R	R	R	Noise				
							I	C	I	Lighting				
							R	R	I	Driving conditions, balance, slipping/falling				
							C	C	I	Journey plan				
							C	C	I	Carrier design				
							C	C	C	Facilities (ramp/race, holding pen)				
							C	R	C	Hygiene of facilities				
Good health	6	Absence of injuries	C	C	I	Mortality								
			C	C	R	Morbidity and health								
	7	Absence of disease	C	C	R	Morbidity and health								
	8	Absence of pain induced by management procedures	R	I	I	Hormones								
			R	I	I	Haematology								
		Other	I	I	I	Reproductive efficacy								
			C	C	R	Pregnant status								
			I	I	R	Meat quality and yield								
Appropriate behaviour	9	Expression of social behaviour	R	R	R	Ethograms								
			R	R	R	Stereotypic behaviour								
			R	R *	R	Emotional state								
	10	Expression of other behaviour	I	I	I	Heart rate								
			I	I	I	Heart rate variability								
	11	Good human-animal relationship	R	R	R	Flight zones								
			R	R	R	QBA								
	12	Positive emotional state	R	R	R	QBA								
Other						**Management-based**				**Industry-management**				**Management-based**
			C	C	I	Appropriate sourcing (incl. breed, genotype, size, age)	C	C	C	License and Accreditation	C	C	C	Stocking rate
											C	I	R	Time at feedlot
			C	C	C	Mixing	C	C	C	Assurance schemes	C	C	C	Euthanasia
			C	C	C	Isolation/separation	C	C	C	Auditing and compliance	I	I	C	Slaughter method
			C	C	C	Hospital pen	C	C	C	Documentation and Reporting				
			R	R	R	Habituation to transport or handling								
							R	C	C	Stock handler skills				
			C	C	C	Traceability	C	C	C	Standard operating procedures				
			C	R	I	Rejections								
			C	R	C	Use of electric prods/dogs/stick etc.								

^a^ Qualitative Behavioural Assessment (QBA). * Indicates a measure that has been suggested to be included in ASEL regulations by the Technical Advisory Committee 2018 [58] but is not currently under ASEL v2.3 (2011) regulations. Current practice (C) were those included in the reviewed legislation and regulations for the LEI, including the proposed revisions by the Technical Advisory Committee 2018 for ASEL sea [59] and 2019 ASEL air transport reporting requirements [60].

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
