# Peer review of "Review of Livestock Welfare Indicators Relevant for the Australian Live Export Industry"

_animals, 2020, doi:10.3390/ani10071236_

Round 1
Reviewer 1 Report
Revision: Review of Livestock Welfare Indicators Relevant for the Australian Live Export Industry
The purpose of this paper is to identify relevant welfare indicators for each point in Australian live export industry (feeder and slaughter cattle and sheep) by reviewing of accepted and used indicators at international level. This could contribute to developing a benchmarking system for animal welfare assessment using a Dashboard.
The transport and export of live animals is the sore point of animal production and animal welfare. The scenario can be very varied from country to country and consequently also the regulations. The policy makers have to take into account different aspects and situations (medium-long and short-range travels; different species; ethological needs…etc.)
So, the development of a system/protocol to monitoring the animal welfare during all stages of export is desirable and necessary.
The paper represents a great job of revising of international and current livestock welfare indicators and of analysis of relevant laws and regulations in force for LEI.
The introduction should be organized more efficiently to be more understandable.
It would be useful to insert a summary at the beginning of the article to facilitate the understanding of the subdivision of the manuscript.
It is quite difficult to understand the organization of the paragraphs and their subparagraphs.
It could be also useful put a list of acronyms
Although this paper is a review, the general indications for scientific article could be taken into account in order to facilitate the reading of the manuscript. For example, explain better the approach used for the review (e.g. searching in database, number of articles reviewed, the selection of articles or regulations/legislations, etc., the criteria of inclusion/exclusion).
In general conclusions are a bit weak and should be improved in order to highlight the main points resulted from review, considering also the aims.
Introduction
BOX 1 the animal copes with the stressors. The coping strategy involve physiological, immune and behavioral response. This response, when is not useful or adaptative and the stress situation is not resolved, may affect the animal welfare. In some case the stressor cannot be considered negative but depend on the animal experience, perception, genetic etc. I think that this aspect should be considered.
You have used the five domains of Mellor et al. plus the Principle of Welfare Quality. Is it correct?
Figure 1 cattle by ship, point 2 do you mean for example rough seas or sinking of ship? I suggest putting some examples.
Figure 1 in some cases the contributing factors overlap mortality causes (Inadequate ventilation).
Figure 1 is Insufficient feed…another point of the list?
Figure 2: it is not very clear the difference between the two figures (1-2)
LL 85-94 this part needs to be reorganized to give a better consequence of the concepts expressed. LL 90-94 should be moved up al LL 85.
LL 129 ESCAS: it should be the first time that is cited in the manuscript. So, should be useful put the explanation of acronym.
LL 143 Are the stages the same cited in LL 28? I suggest adding the explanation of the stages also in brackets (1. Australian facilities, 2. Vessel etc.)
LL 157 delete voyage: > 10 days
LL 161 IOs: the same per ESCAS
LL 182-183 This point should be explained better
LL 184 reference?
LL 209-210 Also this point in not totally clear. The animals respond to a stressor such as loading. This aspect should be taken into account for welfare assessment. I mean, the animal welfare assessment should consider the different situation of value chain: at farm, during loading, during transport, at slaughterhouse etc. It would be desirable to have assessment protocols including relevant welfare measures for each situation or for each stages or steps of LEI. Add also references.
LL 216 reference
LL 232-236 in my opinion this sentence sounds better as a conclusion
LL 238-240 these are the aims of the paper. I suggest harmonizing the aims across the paper in order to underline better the objective of the paper (see Summary and Abstract)
LL 243-245 As said previously, this part could be like materials and methods. It is not totally clear where the 58 potential indicators come from.
LL 247 Does vessel mean? Airplane and ship?
LL 249 Does “each measure” refer to 58 potential indicators?
LL 259 What does behavioral scores refer to? An example?
LL 264 which measures? Examples?
LL 269 and LL 279which measures?
LL 278-279 Does it mean developing a protocol to assess animal welfare at post-voyage discharge phase?
LL 283-285 this part needs a clarification
LL 294-295 a risk analysis?
LL 297 it is important to underline that the concept of animal welfare is not only absence of disease
LL 304 Why unavoidable? Explain better this point
LL 308 poor demeanor: it is not clear, what does it mean?
LL 311 assessment at pen level can mean assessing all animals housed in that pen or a sample of them
LL 337-339 panting or increasing in respiratory rate could be also related to anxiety or fear. So, this measure is not totally reliable and valid for heat stress and thermoregulation.
LL 359 not currently recorded= not undertaken by the LEI?
LL 387-388 I suggest explaining better how the assessment is done for gut fill
LL 409 behavioral that occurs infrequently, what does it mean? Explain better this point because the behavioral assessment is not only for infrequent behavioral events.
LL 413-417 Stereotypical behaviors can be also a coping strategy. I suggest adding this point. Not only boredom can trigger these behaviors, but also stress situation or emotional state of animal (see Mason & Rushen 2008 Stereotypic animal behavior: fundamentals and applications to welfare, Mason and Latham 2004).
LL 417 Ethogram is the set of natural behaviors that manifests a specific animal species in general or in a specific situation. It is not clear in this case what the authors refer to. Please explain better this aspect.
LL 430-422 it is a repetition of LL 415-416
LL 433-434 I suggest deepening better this point
LL 435-438 QBA is used to assess the emotional state of animals (see Welfare Quality). This point needs a better explanation on QBA. A brief introduction on this tool could be useful.
LL 442 explain better what is this informal QBA approach, I am not totally agree with authors’ point of view. QBA is a specific tool with its rules for the assessment. The perception of stockmen on animal behaviors is more an empiric evaluation.
LL 446 I don’t understand well if environmental aspects have been already used in the welfare assessment in LEI.
LL 454-455 are these procedures already implemented during voyage? Or should be carried out?
LL 463 it is not totally clear if the coaction of this information (n of livestock born, n of abortions...?) is already implemented and the authors would like to add in the Welfare Dashboard.
Conclusions
LL 516 Sorry, but I didn’t find the figure 4.
It could be very useful to summarize in a table the identified measures useful for the LEI) already in use and new) and also the dismissed measures. This table permits to recap easily all the information collected in the paper. Maybe the figure 4 covers this point.
LL 530-532 this point seems a repetition of LL 518, but one is the review of regulations and the other one is of literature. This aspect could be explained and managed better.
It would be interesting taking into account the use of sensors’ technology (see precision livestock farming) such as environmental sensors (sensors for measure THI or Co2 level, sensors for animal vocalizations) and/or animal sensors where it is feasible. I know that this technology is expensive, and it is difficult to implement in LEI but think about how this technology could be useful it is perhaps even necessary for the purpose of the project, also in a future perspective.
Author Response
Reviewer 1
Thank you very much for the extremely comprehensive and informative review. It has been of great assistance in improving the manuscript.
The purpose of this paper is to identify relevant welfare indicators for each point in Australian live export industry (feeder and slaughter cattle and sheep) by reviewing of accepted and used indicators at international level. This could contribute to developing a benchmarking system for animal welfare assessment using a Dashboard.
The transport and export of live animals is the sore point of animal production and animal welfare. The scenario can be very varied from country to country and consequently also the regulations. The policy makers have to take into account different aspects and situations (medium-long and short-range travels; different species; ethological needs…etc.)
So, the development of a system/protocol to monitoring the animal welfare during all stages of export is desirable and necessary.
The paper represents a great job of revising of international and current livestock welfare indicators and of analysis of relevant laws and regulations in force for LEI.
Thank you for the constructive and supportive comments.
The introduction should be organized more efficiently to be more understandable.
We have re-ordered some of the paragraphs to make the flow of information clearer.
It would be useful to insert a summary at the beginning of the article to facilitate the understanding of the subdivision of the manuscript. It is quite difficult to understand the organization of the paragraphs and their subparagraphs.
We have added a paragraph summarising the layout of the review.
It could be also useful put a list of acronyms
We have added such a list
Although this paper is a review, the general indications for scientific article could be taken into account in order to facilitate the reading of the manuscript. For example, explain better the approach used for the review (e.g. searching in database, number of articles reviewed, the selection of articles or regulations/legislations, etc., the criteria of inclusion/exclusion).
We have provided details on the documents searched and included in the review.
In general conclusions are a bit weak and should be improved in order to highlight the main points resulted from review, considering also the aims.
We have re-worked the conclusions; moving sections from other parts of the manuscript, adding further description of benchmarking vs assurance schemes, including a section on further work needed.
Introduction
BOX 1 the animal copes with the stressors. The coping strategy involve physiological, immune and behavioral response. This response, when is not useful or adaptative and the stress situation is not resolved, may affect the animal welfare. In some case the stressor cannot be considered negative but depend on the animal experience, perception, genetic etc. I think that this aspect should be considered.
We have added text to this section that hopefully addresses this comment. If we have missed the point, however, then it is because we do not properly understand the comment.
You have used the five domains of Mellor et al. plus the Principle of Welfare Quality. Is it correct?
That is correct. We have included reference to the table columns for clarity.
Figure 1 cattle by ship, point 2 do you mean for example rough seas or sinking of ship? I suggest putting some examples.
Added ‘rough sea conditions’
Figure 1 in some cases the contributing factors overlap mortality causes (Inadequate ventilation).
Reworded ‘Suffocation or temperature extremes’
Figure 1 is Insufficient feed…another point of the list?
Yes, it is. We have added a bullet point here.
Figure 2: it is not very clear the difference between the two figures (1-2)
We have merged the content of Fig. 2 with Fig.
LL 85-94 this part needs to be reorganized to give a better consequence of the concepts expressed. LL 90-94 should be moved up al LL 85.
This section has been reworded to address this concern and LL 90-94 has been moved to LL85.
LL 129 ESCAS: it should be the first time that is cited in the manuscript. So, should be useful put the explanation of acronym.
Added.
LL 143 Are the stages the same cited in LL 28? I suggest adding the explanation of the stages also in brackets (1. Australian facilities, 2. Vessel etc.)
Thanks, added
LL 157 delete voyage: > 10 days
The current regulations only require Daily Voyage Reports to be completed and submitted to the Department for voyages ≥10 voyage days, so this information has remained in text.
LL 161 IOs: the same per ESCAS
The acronym was given the line above.
LL 182-183 This point should be explained better
This sentence has been reworded to make the meaning clearer.
LL 184 reference?
Added
LL 209-210 Also this point in not totally clear. The animals respond to a stressor such as loading. This aspect should be taken into account for welfare assessment. I mean, the animal welfare assessment should consider the different situation of value chain: at farm, during loading, during transport, at slaughterhouse etc. It would be desirable to have assessment protocols including relevant welfare measures for each situation or for each stages or steps of LEI. Add also references.
This section has been reworded to address this concern.
LL 216 reference
Added
LL 232-236 in my opinion this sentence sounds better as a conclusion
Good point. Sentence moved
LL 238-240 these are the aims of the paper. I suggest harmonizing the aims across the paper in order to underline better the objective of the paper (see Summary and Abstract)
We have moved this sentence up to the paragraph describing the scope of the review.
LL 243-245 As said previously, this part could be like materials and methods. It is not totally clear where the 58 potential indicators come from.
A sentence detailing the sources for the 58 measures has been added.
LL 247 Does vessel mean? Airplane and ship?
Means both, ship and aeroplane have been added in brackets
LL 249 Does “each measure” refer to 58 potential indicators?
Yes it does. This has been clarified in the sentence.
LL 259 What does behavioral scores refer to? An example?
This means all behaviour measures, so posture, activity and demeanour has been added in brackets to clarify this.
LL 264 which measures? Examples?
Examples have been provided in brackets and reference to Table 3 has been inserted.
LL 269 and LL 279which measures?
Examples of measures have been added and reference to Table 3.
LL 278-279 Does it mean developing a protocol to assess animal welfare at post-voyage discharge phase?
This has been clarified to mean land transport, in feedlots and abattoirs in the destination country.
LL 283-285 this part needs a clarification
The meaning of this has been clarified.
LL 294-295 a risk analysis?
The potential of this to occur has been stated.
LL 297 it is important to underline that the concept of animal welfare is not only absence of disease
This sentence has been amended to reflect this concern.
LL 304 Why unavoidable? Explain better this point
This has been clarified.
LL 308 poor demeanor: it is not clear, what does it mean?
Examples added.
LL 311 assessment at pen level can mean assessing all animals housed in that pen or a sample of them
We understand this. When we refer to measurements at a pen level, it means that all animals within the pen are taken into account for the measure.
LL 337-339 panting or increasing in respiratory rate could be also related to anxiety or fear. So, this measure is not totally reliable and valid for heat stress and thermoregulation.
As this measure would not be taken in isolation, we have added that the inclusion of environmental measures would be able to determine the cause of panting… i.e heat stress or fear response.
LL 359 not currently recorded= not undertaken by the LEI?
Yes, we mean that this section addresses measures not recorded by the LEI. We have amended the section title to reflect this.
LL 387-388 I suggest explaining better how the assessment is done for gut fill
Clarification of this has been added.
LL 409 behavioral that occurs infrequently, what does it mean? Explain better this point because the behavioral assessment is not only for infrequent behavioral events.
This has been modified to be an introductory sentence for the section and includes frequent and infrequent behaviours.
LL 413-417 Stereotypical behaviors can be also a coping strategy. I suggest adding this point. Not only boredom can trigger these behaviors, but also stress situation or emotional state of animal (see Mason & Rushen 2008 Stereotypic animal behavior: fundamentals and applications to welfare, Mason and Latham 2004).
Point taken. We have included this comment, thanks.
LL 417 Ethogram is the set of natural behaviors that manifests a specific animal species in general or in a specific situation. It is not clear in this case what the authors refer to. Please explain better this aspect.
Reworded – ‘activity budgets’ instead of ethograms
LL 430-422 it is a repetition of LL 415-416
We have clarified to ensure that the differences between activity budgets and preferences are more obvious.
LL 433-434 I suggest deepening better this point
We have added further explanation of QBA.
LL 435-438 QBA is used to assess the emotional state of animals (see Welfare Quality). This point needs a better explanation on QBA. A brief introduction on this tool could be useful.
We have added further explanation of QBA.
LL 442 explain better what is this informal QBA approach, I am not totally agree with authors’ point of view. QBA is a specific tool with its rules for the assessment. The perception of stockmen on animal behaviors is more an empiric evaluation.
Section reworded
LL 446 I don’t understand well if environmental aspects have been already used in the welfare assessment in LEI.
An introduction sentence covering this has been added to this subsection.
LL 454-455 are these procedures already implemented during voyage? Or should be carried out?
This is already regulated and managed under the LEI, with this information added.
LL 463 it is not totally clear if the coaction of this information (n of livestock born, n of abortions...?) is already implemented and the authors would like to add in the Welfare Dashboard.
This has been clarified.
Conclusions
LL 516 Sorry, but I didn’t find the figure 4.
Apologies – that was a late deletion that had not been picked up.
It could be very useful to summarize in a table the identified measures useful for the LEI) already in use and new) and also the dismissed measures. This table permits to recap easily all the information collected in the paper. Maybe the figure 4 covers this point.
Table 3 captures these data, including where measures are already in use and new, or dismissed. If we summarise the data from table 3 further, then we will compromise the value of having the table.
Apologies –removal of Figure 4 was a late deletion that had not been picked up.
LL 530-532 this point seems a repetition of LL 518, but one is the review of regulations and the other one is of literature. This aspect could be explained and managed better.
Thanks for picking this up – we have re-worded this section.
It would be interesting taking into account the use of sensors’ technology (see precision livestock farming) such as environmental sensors (sensors for measure THI or Co2 level, sensors for animal vocalizations) and/or animal sensors where it is feasible. I know that this technology is expensive, and it is difficult to implement in LEI but think about how this technology could be useful it is perhaps even necessary for the purpose of the project, also in a future perspective.
We have added some comments on this.

Reviewer 2 Report
General comments:
This is a well written and researched manuscript in an important research space. I only have a few comments and suggestions for improving the manuscript.
Introduction could perhaps explain public concerns more specifically with live export, is it just mortality? The impact of public perception of adequate welfare is important to the social license and so it is critical to know whether public concerns and expectations align with scientific and industry measures and standards.
Another aspect which may be useful to consider in the context of welfare is the role of predictability and controllability in assessing and addressing welfare issues. Are there any ways in which these factors could be utilised in improving management of animals? Are there ways in which lack of predictability and controllability could be contributing to welfare issues, and could the identification of these play a role in the development of welfare benchmarks?
Thank you for the opportunity to review this manuscript, I look forward to seeing it in print.
Recommended corrections in text:
Line 74: The referenced Livecorp report states almost 10 000 people employed in the cattle industry (page 4), perhaps clarify that this is an approximate number for the live cattle supply chain in the text as it is a bit unclear whether this includes sheep etc in the text. Ensure referenced data is accurate and clear as there is potential for confusion as to the numbers of people employed across the live export sector for all species.
Figure 3: More detail in the figure description is needed, as some aspects are not clear, eg FOB doesn’t appear in the text and is not explained in the figure, perhaps something like “Figure 3. Animal numbers for live export by species (sheep, cattle, goats) and transport (air or sea), and total Free on Board (FOB).”
Lines 150, 156, 164, 176, 241, 265, 286, 307, 345, 441, 444, supplementary material: Please use more inclusive language (e.g. instead of stockmen/man, use stockperson, stock handlers, etc). Stockmanship may be unavoidable but see if there is a better term for stock handling skills that is more inclusive.
Section 2.2 Individual vs group assessments: maybe add some comments that future advances in remote monitoring technologies (temperature sensing using microchip sensors or IRT, automated video analysis etc) may enable better individual assessments in coming years but are not in commercial use at this stage, it may be worth allowing some flexibility in the developing of the framework for these technologies to be utilised as they become available.
Line 264: You talk about low-stress handling and experience etc, may be useful to put in some reference earlier in the text about the impact of these on welfare outcomes.
Author Response
Reviewer 2
General comments:
This is a well written and researched manuscript in an important research space. I only have a few comments and suggestions for improving the manuscript.
Introduction could perhaps explain public concerns more specifically with live export, is it just mortality? The impact of public perception of adequate welfare is important to the social license and so it is critical to know whether public concerns and expectations align with scientific and industry measures and standards.
Section reworded.
Another aspect which may be useful to consider in the context of welfare is the role of predictability and controllability in assessing and addressing welfare issues. Are there any ways in which these factors could be utilised in improving management of animals? Are there ways in which lack of predictability and controllability could be contributing to welfare issues, and could the identification of these play a role in the development of welfare benchmarks?
this comment is addressed in lines 106-111: “The issue with simply monitoring mortality and non-compliance with regulation is that problems can only be indicated retrospectively. Under current animal welfare reporting, the main welfare issues identified for the LEI are environmental conditions, resource access, and species-specific causes of morbidity and mortality (Figure 2). Being able to effect change proactively, on the basis of animal outcomes that do not result in mortality, could enable pre-emptive modifications and adaptive management solutions before animal welfare is compromised”.
Thank you for the opportunity to review this manuscript, I look forward to seeing it in print.
Recommended corrections in text:
Line 74: The referenced Livecorp report states almost 10 000 people employed in the cattle industry (page 4), perhaps clarify that this is an approximate number for the live cattle supply chain in the text as it is a bit unclear whether this includes sheep etc in the text. Ensure referenced data is accurate and clear as there is potential for confusion as to the numbers of people employed across the live export sector for all species.
Reworded
Figure 3: More detail in the figure description is needed, as some aspects are not clear, eg FOB doesn’t appear in the text and is not explained in the figure, perhaps something like “Figure 3. Animal numbers for live export by species (sheep, cattle, goats) and transport (air or sea), and total Free on Board (FOB).”
Thanks – the legend has been updated.
Lines 150, 156, 164, 176, 241, 265, 286, 307, 345, 441, 444, supplementary material: Please use more inclusive language (e.g. instead of stockmen/man, use stockperson, stock handlers, etc). Stockmanship may be unavoidable but see if there is a better term for stock handling skills that is more inclusive.
Thanks for picking this up. We have changed throughout.
Section 2.2 Individual vs group assessments: maybe add some comments that future advances in remote monitoring technologies (temperature sensing using microchip sensors or IRT, automated video analysis etc) may enable better individual assessments in coming years but are not in commercial use at this stage, it may be worth allowing some flexibility in the developing of the framework for these technologies to be utilised as they become available.
The development of automated systems for the assessment of animal welfare is an extensive field of research on its own. It would be problematic to introduce statements alluding to devices the validity or representativeness have yet to be tested.
Line 264: You talk about low-stress handling and experience etc, may be useful to put in some reference earlier in the text about the impact of these on welfare outcomes.
References added
